



**Spatiotemporal transformation of dissolved organic matter along an alpine stream**
**flowpath on the Qinghai-Tibetan Plateau: importance of source and permafrost**
**degradation**
Yinghui Wang [a,b], Robert G.M. Spencer [c], David Podgorski [d], Anne Kellerman [c],
Harunur Rashid [a], Phoebe Zito [d], Wenjie Xiao [b], Dandan Wei [a], Yuanhe Yang [e], Yunping
Xu [a*]
[a] *Shanghai Engineering Research Center of Hadal Science and Technology, College of Marine*
*Sciences, Shanghai Ocean University, Shanghai 201306, China.*
[b] *Key Laboratory for Earth Surface Processes of the Ministry of Education, College of Urban and*
*Environmental Sciences, Peking University, Beijing 100871, China.*
[c] *National High Magnetic Field Laboratory Geochemistry Group and Department of Earth, Ocean,*
*and Atmospheric Science, Florida State University, Tallahassee, FL 32306, USA*
[d] *Pontchartrain Institute for Environmental Sciences, Department of Chemistry, University of New*
*Orleans, New Orleans, LA, 70148, USA*
[e] *State Key Laboratory of Vegetation and Environmental Change, Institute of Botany, Chinese*
*Academy of Sciences, Beijing 100093, China*
*Corresponding author. *E-mail*: ypxu@shou.edu.cn (Y. Xu)



**Abstract** The Qinghai-Tibetan Plateau (QTP) accounts for approximately 70% of
global alpine permafrost and is an area sensitive to climate change. The thawing and
mobilization of ice and organic carbon-rich permafrost impact hydrologic conditions
and biogeochemical processes on the QTP. Despite numerous studies of Arctic
permafrost, there are no reports to date for the molecular-level in-stream processing of
permafrost-derived dissolved organic matter (DOM) on the QTP. In this study, we
examine temporal and spatial changes of chemical composition of DOM and $^{14}$C age
of dissolved organic carbon (DOC) along an alpine stream (3850–3207 m above sea
level) by Fourier transform ion cyclotron resonance mass spectrometry (FT-ICR MS),
accelerator mass spectrometry (AMS) and UV-visible spectroscopy. Compared to
downstream sites, the DOM at the headstream exhibited older radiocarbon ($^{14}$C-DOC)
age, higher mean molecular weight, higher aromaticity and fewer polyunsaturated
components. At the molecular level, 6409 and 1345 formulas were identified as unique
to the active layer (AL) leachate and permafrost layer (PL) leachate, respectively.
Comparing permafrost leachates to the downstream site, 59% of AL-specific formulas
and 90% of PL-specific formulas were degraded, likely a result of rapid instream
degradation of permafrost-derived DOM. From peak discharge in the summer to low
flow in late autumn, the DOC concentration at the headstream site decreased from 13.9
to 10.2 mg/L, while the $^{14}$C-DOC age increased from 745 to 1560 years before present
(BP), reflecting an increase in relative contribution of deep permafrost carbon due to
the effect of changing hydrological conditions over the course of the summer on DOM
source (AL vs. PL). Our study thus demonstrates that hydrological conditions impact
the mobilization of permafrost carbon in an alpine fluvial network, the signature of
which is quickly lost through in-stream metabolism.
Keywords: dissolved organic matter; permafrost; Qinghai-Tibet Plateau; FT-ICR MS;



radiocarbon age

## 1.   INTRODUCTION

The amount of carbon stored in permafrost is roughly twice as much as that in the
atmosphere and represents the largest component of the terrestrial carbon pool (Zimov
et al., 2006; Tarnocai et al., 2009). Accelerated climate warming has led to a succession
of changes associated with permafrost thaw, where liquid water once frozen in
permafrost soils has changed watershed hydrology, topography and ecosystem
biogeochemistry (Frey and Smith, 2005; Abbott et al., 2015; Vonk et al., 2015). When
permafrost-derived carbon enters aquatic systems, it can be rapidly bio- and photo-
degraded (Cory et al., 2014; Drake et al., 2015; Vonk et al., 2015). Therefore, the
mobilization of carbon from permafrost soils where it has been relatively stable for
thousands of years into dissolved carbon could increase greenhouse gas emissions
(Cory et al., 2013; Vonk et al., 2013; Mann et al., 2015; Ward and Cory, 2016; Selvam
et al., 2017) and exacerbate climate warming via a positive feedback loop (Koven et al.,
2011; Schuur et al., 2015).
The seasonal thawing-freezing cycle of permafrost soils could change hydrologic
inputs and restrict source water contributions to river flow, leading to variability in the
flux and the chemical composition of dissolved organic matter (DOM) in permafrost-
impacted watersheds (Petrone et al., 2006; Laudon et al., 2011). DOM in the Yukon
River exhibits seasonal changes in aromaticity, molecular weight, $^{14}$C age and
biodegradability (Striegl et al., 2007; Spencer et al., 2008; Wickland et al., 2012;
O'Donnell et al., 2014). Since the persistence of DOM in aquatic systems is related to
chemical composition (Kellerman et al., 2015; Kellerman et al., 2018), substituting
space for time to trace changes in DOM composition along a hydrologic flowpath may



illustrate the environmental behavior and fate of seasonally exported permafrost carbon.

The Qinghai-Tibet Plateau (QTP), the world's largest and highest plateau, plays a

critical role in the evolution of the Asian Monsoon (Sato and Kimura, 2007; Wu et al.,
2007) and supplies water to several large rivers such as the Yangtze River, Yellow River
and Yarlung Tsangpo (Yao et al., 2007; Kang et al., 2010). As a climate sensitive region,
the QTP has experienced significant warming since the 1950s (Qiu, 2008) with the
mean annual air temperature rising at a rate of 0.36 °C per decade from 1961 to 2007
(Wang et al., 2008). Consequently, the permafrost soils on the QTP have begun to thaw
and collapse, causing abundant carbon loss from degradation, leaching and lateral
displacement (Mu et al., 2016). However, compared with an abundance of studies on
Arctic permafrost, biogeochemical studies on QTP permafrost are scant (Mu et al.,
2016). This results in a limited understanding of the permafrost carbon cycle as a whole
since the QTP represents nearly 10% of the global permafrost, what's more, the QTP
differs from the Arctic in altitude, climate, and hydrology (Bockheim and Munroe,

2014).

Here, we conducted a study on the spatial and temporal change of permafrost-derived

DOM on the northeastern QTP. We applied multiple analytical techniques including
Fourier transform ion cyclotron resonance mass spectrometry (FT-ICR MS), AMS
radiocarbon ($^{14}$C), and UV-visible optical spectroscopy. Our objective is two-fold: 1)
determine the dominant sources of alpine stream DOM on the QTP (active layer (AL)
vs. permafrost layer (PL)), and 2) trace the persistence and degradation of permafrost-
derived DOM in an alpine fluvial network. This work represents the first step in
characterizing in-stream removal and transformation processes of permafrost carbon at
the molecular level on the QTP.



## 2. MATERIALS AND METHODS

### 2.1. Study area and sampling

Our study area is located in Gangcha County, north of Qinghai Lake. The climate is typical plateau continental climate, characterized by extensive sunshine duration (~3000 hours per year), long cold and dry winters and short cool and humid summers (Peng et al., 2015). During 2013-2016, January had the lowest average monthly temperature ($-11.82$ °C), while December had the lowest average monthly precipitation (0.3 mm). Meanwhile, the highest average monthly temperature and precipitation occurred in July (11.66 °C) and August (124.67 mm), respectively. These climate data are available at http://data.cma.cn. The permafrost soil was developed in the late Quaternary, and accumulated > 2 m thick in mountainous areas of the Gangcha County. Due to the rapid climate warming on the QTP, the ice-rich permafrost began to thaw, and several thermo-erosion gullies formed a decade ago. In this study, we focused on a continuous system that starts with a thermo-erosion gully (> 200 m long), forms a stream which flows into Qinghai Lake, the largest lake in China with a surface area of ca. 4500 km$^2$. Thawed permafrost slumping exposed soil profiles at the gullies' head (ca. 3850 meters above sea level; masl). The top 60 cm is an active layer (AL) that comprises abundant grass litter and roots, underlain by a dark permafrost layer (PL) without visible plant debris. The thaw depth reached 78 cm in August 2015. Seasonal thaw of the entire AL and the upper PL allows for both vertical and lateral percolation of rainwater, which mobilizes large amounts of particulate and dissolved organic matter. The water in the gully flows southward across the hillslope before draining into Qinghai Lake (3196 masl; Fig. 1).

Our fieldwork was conducted in the summer and autumn of 2015 and 2016. In 2015, a time-series sampling campaign was conducted at the headstream (Q-1) from August





1st when the warm and humid climate caused the largest export of leachates, to October
15th when the leaching ceased due to little precipitation and low temperature. The AL
and PL leachate samples were collected at a depth of 60 cm and 220 cm, respectively,
at the gullies' head. For each leachate sample, >15 L water was gathered over 2 days
using a 20 L pre-cleaned HDPE carboy. Besides soil leachates, water samples (20 L
each) were collected from twenty sites along the stream (Fig. 1). Sampling sites Q-1 to
Q-10 are located in a first-order stream that originates in the largest thermo-erosion
gully, whereas sites Q-11 and Q-12 are located in another first-order stream nearby.
These two streams merge together to form the main stream, along which sampling sites
Q-13 to Q-20 were located. Surface water samples were collected using pre-cleaned
HDPE carboys and kept on ice and in the dark until filtering through Whatman GF/F
filters (0.7 μm) within 6 hours after sampling. To obtain enough carbon for $^{14}$C analyses,
aliquots of the 0.7 μm filtrate were concentrated over a cross-flow ultrafiltration system
with 1 kDa cut off (Millipore®, Pellicon 2 system). The retentates and the remaining
filtrate were all stored at –20 °C until further analysis. All glassware and GF/F filters
were combusted at 450 °C for at least 4 hours prior to use.

**2.2. Hydrological condition, DOC concentration and spectral absorbance in alpine**
**streams**
On Aug. 1st 2015, stream water temperature, pH and conductivity were measured
with a portable Horiba W-23XD Water Quality Monitoring System. The water flux was
calculated according to flow rate and cross-sectional area of the stream. The DOC
concentration of each water sample was determined by 3-5 injections on a Shimadzu
TOC-V$_{CPH}$ analyzer using high temperature combustion, and the coefficient of variance
across measurements was < 2%.





The optical properties of the water samples were determined using a Shimadzu UV-
1800 spectrophotometer. The scan range was between 200 and 800 nm and Milli-Q
water (18.2 M$\Omega$ cm$^{-1}$) was used as the blank. A quartz cell with 1.0 cm path length was
used. The spectral slope of the 275–295 nm region ($S_{275-295}$), an indicator for the
molecular weight of DOM (Helms et al., 2008), was determined by applying log linear
fits across the wavelengths 275-295 nm. Specific UV absorbance (SUVA$_{254}$), an
indicator for relative aromatic C content, was calculated by dividing the decadic UV
absorbance at 254 nm with DOC concentration (Weishaar et al., 2003).

**2.3. Electrospray ionization Fourier transform ion cyclotron resonance mass**
**spectrometry (ESI FT-ICR MS)**
Selected water samples collected in 2016 from headstream (Q-1), mid-stream (Q-9),
and downstream (Q-17), as well as leachate samples collected from the AL and PL,
were prepared for FT-ICR MS analyses. They were solid phase extracted (SPE) using
the Bond Elut PPL (Agilent Technologies) following the procedures of Dittmar et al.
(2008). The aliquot volume of SPE DOM was adjusted for a target final eluate
concentration of 40 μg C/ml (in methanol) to aid ionization in negative mode
electrospray ionization (ESI). The methanol extracts were analyzed on a 9.4 Tesla
custom-built FT-ICR MS at the National High Magnetic Field Laboratory (NHMFL;
Tallahassee, FL; Kaiser et al., 2011). The injection speed was 0.7 μL/min. A total of 100
broadband scans was accumulated for each mass spectra. Other instrumental parameters
can be found in Hodgkins et al. (2016). After internal calibration in MIDAS Predator
Analysis (NHMFL), formulas were assigned based on published rules to peaks with
intensities > 6σ baseline noise (Stubbins et al., 2010) using EnviroOrg®™ software and
categorized by compound class based on the elemental composition of molecular



formulas (Spencer et al., 2014; Corilo, 2015). A modified aromaticity index (AI$_{mod}$) was
calculated according to the definition of Koch and Dittmar (2006): AI$_{mod}$ =
$\frac{1+C-0.5O-S-0.5H}{C-0.5O-S-N}$, and if AI$_{mod}$ is negative, then AI=0. The groups referenced in this study
are: 1) aliphatics (Ali.): H/C 1.5 - 2.0, O/C < 0.9, N = 0; 2) peptides (Pep.): H/C 1.5 -
2.0, O/C < 0.9, N > 0; 3) highly unsaturated compounds (Uns.): AI$_{mod}$ < 0.5, H/C < 1.5;
4) polyphenols (Pol.): 0.5 < AI$_{mod}$ < 0.67; 5) condensed aromatics (CA): AI$_{mod}$ ≥ 0.67.
The relative abundance of the defined compound class was weighted by signal
magnitude in each spectrum.

**2.4. Radiocarbon analyses**
Freeze-dried retentates were fumigated with concentrated hydrochloric acid (12 M)
in order to remove inorganic carbon. After that, the samples were analyzed on the Keck
Carbon Cycle Accelerator Mass Spectrometry (AMS) Facility at the University of
California, Irvine, USA. Processing blank and sample preparation backgrounds were
subtracted. Radiocarbon concentrations are given as conventional $^{14}$C age following
Stuiver and Reimer (1993).

**3. RESULTS**
**3.1. Hydrology and DOC concentration from headstream to downstream water**
Discharge increased along the stream reach, from 0.15 m$^3$/min at the headstream
(Q-1) on August 1$^{st}$ 2015 to 24.14 m$^3$ /min (Q-19) (Fig. 2). pH increased from 7.4 at Q-
1 to 8.2 at Q-4 and remained elevated in the middle and lower stream (7.9 to 8.4).
Conductivity was relatively constant from Q-1 to Q-6 (35 to 38 μs/cm), then increased
at Q-7 and remained elevated downstream (48 to 60 μs/cm). The DOC concentration
was high in headstream waters (e.g., 11.7 ± 0.9 mg/L at Q-1 and 10.2 ± 1.5 mg/L at Q-





2; mean ± SD, same hereafter) and decreased downstream (2.5 to 5.8 mg/L from Q-5
to Q-20). The mean DOC concentration of the AL leachates (11.6 ± 1.1 mg/L) was an
order of magnitude lower than that of the PL leachates (126.4 ± 20.9 mg/L).

**3.2. Optical properties of DOM in leachates and stream waters**

The mean $S_{275-295}$ was (14.5 ± 0.48) × $10^{-3}$ $nm^{-1}$ for the AL leachates and (18.3 ±

1.3) × $10^{-3}$ $nm^{-1}$ for the PL leachates. In the stream waters, the $S_{275-295}$ ranged from 15.8×
$10^{-3}$ to 22.5 × $10^{-3}$ $nm^{-1}$, increasing in downstream reaches. $SUVA_{254}$ was 3.52 ± 0.24 L
mg $C^{-1}$ $m^{-1}$ for the AL leachates and 0.95 ± 0.14 L mg $C^{-1}$ $m^{-1}$ for the PL leachates, and
decreased in the stream from Q-1 to Q-10 (3.06 to 1.27 L mg $C^{-1}$ $m^{-1}$), and then remained
low (Fig. 3). A strong negative correlation was found between $SUVA_{254}$ and $S_{275-295}$ for
water samples from both years ($R^2$ = 0.77, $P$ < 0.01). Neither stream waters nor
permafrost leachates show an interannual variation of optical properties (Fig. 3).

**3.3. Spatiotemporal change of $^{14}$C-DOC age through fluvial networks**

$^{14}$C-DOC age of the PL leachate was 4145 years BP, which was much older than

that of the AL leachate (535 years BP; Fig. 4a). The $^{14}$C-DOC age decreased along the
stream from 745 years BP for the headstream water (Q-1) to 160 years BP at Q-19, a
site close to Qinghai Lake. Besides apparent spatial variability, the $^{14}$C-DOC age also
changed temporally. In 2015, the $^{14}$C-DOC age of the headstream water (Q-1) increased
from 745 years BP on August 1$^{st}$, to 1015 years BP on August 11$^{th}$ and 1560 years BP
on September 5$^{th}$ (Fig. 4b).

**3.4. FT-ICR MS characterization of SPE-DOM**

Compared with the PL leachate, the AL leachate was characterized by higher





molecular richness (14709 vs. 9645 assigned formulae), higher mean molecular weight
(498.81 vs. 452.73 Da) and higher $AI_{mod}$ (0.47 vs. 0.30). Elemental composition
revealed that compounds containing both N and S were only detected in the AL
leachates and headstream waters. The AL leachate contained 54.28% highly unsaturated
compounds, 27.10% polyphenols and 17.23% condensed aromatic compounds,
whereas the proportion of aliphatics and peptides was minor (ca. 1.30%). Compared
with the AL leachate, the PL leachate comprised a higher proportion of polyunsaturated
compounds (74.23%) and aliphatics + peptides (10.04%), but a lower proportion of
polyphenols (11.33%) and condensed aromatics (4.32%).

Along the stream (Q-1, Q-9, and Q-17), the molecular richness, mean molecular
weight and modified aromaticity index of SPE-DOM decreased by 26% (14924 to
11074), 4.7% (510.1 to 486.5 Da), and 16.3% (0.43 to 0.36), respectively (Table 1). The
relative abundance of aromatics (condensed aromatics and polyphenols) decreased by
48% (35.7% at Q-1 vs. 18.4% at Q-17), whereas that of highly unsaturated compounds
increased by 28% (62.8% at Q-1 vs. 80.3% at Q-17). Aliphatics and peptides were
minor components of stream DOM (<1.3%) and did not exhibit a downstream trend.

**4. DISSCUSSION**
**4.1. AL leachates as a major source of stream DOM**
The UV-visible optical parameters and molecular formulas resolved by FT-ICR MS
show that the AL and PL leachates have different chemical compositions (Table 1 and
2). Since chemical composition impacts the reactivity of DOM (Kellerman et al., 2015),
the differing chemical composition between the AL and PL leachates that enter the
stream may influence bioavailability (Vonk et al., 2013) and photolability (Stubbins et
al., 2017). Thus, distinguishing DOM source is crucial for understanding in-stream





biogeochemical processes in permafrost-impacted systems. DOM may originate from
a variety of sources including permafrost soil (AL and PL) leaching, in-situ microbial
production, and wet deposition from snow and rain. At the headstream site (Q-1),
however, the dominant source of DOM is permafrost soil leaching, as short residence
times at the gully head restrict in-stream production and wet deposition is likely
negligible due to low DOC concentrations in Tibetan glaciers (0.2-3.3 µg/ml; Spencer
et al., 2014). Assuming that headstream DOM is derived only from permafrost soil
leaching, we are able to estimate the relative contributions of DOM from the AL and
PL.
The DOC concentration of the AL leachate (ca. $11.6 \pm 1.1$ mg/L; mean $\pm$ SD based
on samples from 2015 and 2016, n = 2; same hereafter) is similar to that of the
headstream (Q-1; ca. $11.7 \pm 0.9$ mg/L), but substantially lower than that of the PL
leachates (ca. $126.4 \pm 20.9$ mg/L), supporting a predominance of AL-leachate DOM in
stream waters. In addition, the $SUVA_{254}$ is $3.51 \pm 0.24$ L mg $C^{-1}$ $m^{-1}$ for AL leachates
(and $0.95 \pm 0.14$ L mg $C^{-1}$ $m^{-1}$ for PL leachates, whereas the $S_{275\text{-}295}$ is $(14.5 \pm 0.48) \times$
$10^{-3}$ $nm^{-1}$ for AL leachates and $(18.0 \pm 1.33) \times 10^{-3}$ $nm^{-1}$ for PL leachates. Similar optical
properties and DOC concentrations between AL-leachates and the headstream water
$(16.5 \pm 0.40 \times 10^{-3}$ $nm^{-1}$ for $S_{275\text{-}295}$ and $2.92 \pm 0.19$ L mg $C^{-1}$ $m^{-1}$ for $SUVA_{254}$) support
DOM that leaches from the AL dominates stream DOM. Furthermore, the stream water
at Q-1 has a $^{14}$C-DOC age of 745 years BP, close to that of the AL leachate (535 years
BP), and much younger than that of the PL leachate (4145 years BP). We estimate the
portion of AL and PL-derived C by using a binary mixing model based on $\Delta^{14}$C values
of bulk DOC (Criss, 1999):
$\Delta^{14}C_{DOM} = f_{AL} \times \Delta^{14}C_{AL} + f_{PL} \times \Delta^{14}C_{PL}$
$1.0 = f_{AL} + f_{PL}$



According to this model, ca. 94% of DOC collected from stream site Q-1 on Aug.
1, 2015 originated from the AL (Fig. 6a). Headstream $^{14}$C-DOC age increased from
summer to fall (Fig. 4b), reflecting an enhanced contribution of old carbon from the
deeper soils (i.e., PL), however, the AL still accounted for $\geq$ 72% of total DOC exported
(Fig. 6a). This binary mixing model may overestimate the contribution of AL to stream
DOC since PL-derived DOC may be degraded faster than AL-derived DOC, due to the
high biolability of ancient permafrost carbon as shown in Arctic ecosystems (Vonk et
al., 2013). Nonetheless, the AL appears as a major contributor to stream DOC in the
QTP.
Seasonal variation of $^{14}$C-DOC (Fig. 4b) has been previously observed in high
latitude permafrost areas in Alaska (Aiken et al., 2014; O'Donnell et al., 2014), with the
most enriched $^{14}$C values observed in the spring and becoming more depleted through
summer-fall and/or during winter. The mean monthly air temperature of Gangcha
County, after reaching the maximum in July (ca. 10.5 ºC), decreases to 2.1 ºC in
September (data from http://data.cma.cn). As air temperature drops, surface soils freeze
earlier than deeper soils, leading to an increase in the relative contribution of deep soil
carbon (i.e. PL) to stream DOM, although the DOC concentration in Q-1 decreased
from 13.9 mg/L to 10.2 mg/L (Fig. 6b).

**4.2. Selective removal of DOM along the alpine stream on the QTP**
The DOC concentration decreased (12.3 to 4.0 mg/L) from upper to mid-stream
(Q-1 to Q-5), which could be attributed to a dilution effect and/or in-stream degradation
of DOM. Dilution from groundwater is likely since groundwater discharge sustains
baseflow of rivers and streams in the QTP (Ge et al., 2008). Downstream groundwater
inputs were further supported by the order of magnitude increase in discharge (1.49 to





24.14 m$^3$/min) and increase in conductivity (37 to 60 µs/cm). Moreover, downstream
DOC concentrations remained about 3.0-4.0 mg/L (Q-15 to Q-20), indicative of the low
DOC concentrations of groundwater. Conversely, a tributary that originated from
another thermo-erosion gully merged into the study stream, however, the different
tributaries exhibited similar DOC concentrations (e.g., Q-9 and Q-10 vs. Q-11 and Q-
12; Fig. 2d). The similarities in DOC concentrations were attributed to homogeneity in
dominant vegetation, soil type and climate, and thus, homogeneity in DOM inputs to
the different tributaries in our study area. Therefore, additional tributaries could not
explain the spatial pattern of DOC concentration.
Despite evident dilution, DOC attenuation could be partly due to in-stream
degradation given several lines of evidence from optical properties, radiocarbon age
and molecular composition. The UV-visible optical parameters, $S_{275-295}$ and SUVA$_{254}$,
have been widely used to reveal mean molecular weight and aromaticity of DOM,
respectively (Weishaar et al., 2003; Helms et al., 2008; Spencer et al., 2009; Mann et
al., 2012). In our study, the $S_{275-295}$ of stream waters varied from $15.8 \times 10^{-3}$ to $22.5 \times$
$10^{-3}$ nm$^{-1}$ (Fig. 3a), comparable to typical riverine DOM values ($13.19 \times 10^{-3}$ to 22.96
$\times 10^{-3}$ nm$^{-1}$), but much lower than that of DOM from continental shelf and slope (29.7
$\times 10^{-3}$ to $48.5 \times 10^{-3}$ nm$^{-1}$) (Fichot and Benner, 2012), suggesting a moderate degradation
of stream DOM on the QTP. A downstream increase for $S_{275-295}$ regardless of sampling
time (Fig. 3a) reflects selected degradation of high molecular weight compounds,
leading to the enrichment of low molecular weight DOM. This spatial trend is in
accordance with the size-reactivity continuum model (Amon and Benner, 1996) that the
bioreactivity of DOM decreases along a continuum of size (from large to small). In
addition to $S_{275-295}$, SUVA$_{254}$ varied from 1.27 to 3.06 L mg C$^{-1}$ m$^{-1}$, showing a general
decrease downstream (Fig. 3b). Lignin, an aromatic biopolymer specific for vascular



plants (Hedges et al., 1997), is relatively resistant to biodegradation (Hedges et al.,
1985), but highly photo-labile (Lanzalunga and Bietti, 2000). Cory et al. (2014) found
that sunlight accounts for 70% to 95% of water column carbon processing in Arctic
rivers and lakes. Given strong solar radiation and long sunshine duration (~3000 hours
per year) on the QTP (Peng et al., 2015), photo-degradation could be an important
pathway for carbon removal in QTP streams. A strong negative correlation between
$S_{275-295}$ and SUVA$_{254}$ ($R^2$ = 0.73, $p$ < 0.01) indicates that photodegradation of high
molecular weight aromatic compounds (like lignin) may play a role in the decrease of
mean molecular weight of DOM along the stream.
Similar to SUVA$_{254}$ and $S_{275-295}$, the data from FT-ICR MS also show a downstream
decrease in aromaticity (AI$_{mod}$: 0.43 to 0.36) and mean molecular weight of stream
DOM (510.0 to 486.5 Da; Table 1). Compared with headstream DOM at Q-1, DOM at
Q-9 and Q-17 was characterized by a lower proportion of condensed aromatics and
polyphenols and enriched in highly unsaturated compounds (Table 1). The decrease in
relative abundance of aromatic compounds is consistent with the reports for the
photolability of aromatic formulas within permafrost, river and ocean DOM (Stubbins
and Dittmar, 2015; Stubbins et al., 2017).
As discussed in section 4.1, AL is the principal contributor to stream DOM. Thus,
tracing AL-derived DOM is paramount in estimating biogeochemical processes of
carbon in the stream. FT-ICR MS identified 6409 molecular formulas specific to AL-
leachates (i.e. not observed in the PL, Table 2). Through various stream processes, some
AL specific formulas were removed from the DOM pool (from 17% by Q-1 up to 59%
by Q-17), which accounted for 66% of the aromatic compounds and 51% of the highly
unsaturated compounds (Table 2). Molecular formulas containing N and/or S were more
labile in the fluvial networks than CHO formulas, with 84% of S-containing formulas



and 100% of S and N-containing formulas lost (Table 2). Furthermore, the removal of
DOM formulas (ca. 83% of AL-specific formulas, and >95% of AL-specific formulas)
occurred in upper and mid-stream (leachates to Q-9). Concurrent with the rapid loss of
AL-specific formulas, some new molecular formulas were detected by FT-ICR MS,
which was mainly attributed to in-situ production by stream algae/microbes, although
an import form groundwater could not be excluded. The addition of those new
molecular formulas was also reflected by the $^{14}C$ enrichment in middle and lower-
stream (Fig. 3b).
Overall, our multiple analyses demonstrate a rapid and selective degradation of
stream DOM on the QTP. The attenuation of aromatic compounds and enrichment of
highly unsaturated compounds could change the environmental photo- and bio-lability
of DOM, increasing relative importance of photodegradation in upper stream and
biodegradation in lower stream. The continuous change in chemical properties of DOM
along the alpine stream flowpath has a potential to shift the aquatic microbial
community since DOM serves as an important energy and nutrient source(Wild et al.,

2014).


**4.3. Prediction of in stream carbon dynamic under continued warming**
The DOC concentrations, UV-visible optical parameters and FT-ICR MS all suggest
that currently, PL is a minor source to stream DOM (see 4.1). However, the QTP is a
sensitive area to global warming, with a rate of air temperature rise that is
approximately three times the global warming rate (Qiu, 2008). According to climate
model predictions, spatial average temperatures of the QTP will increase by 0.68–0.98
ºC for the period of 2015–2050 (Zhu et al., 2013), and in 2050, the mean AL thickness
on the QTP permafrost will increase by approximately 0.3-0.8 m more than that in 2010




(Zhang and Wu, 2012). With the deepening of the AL, carbon that is currently stable in
the PL will be thawed and mobilized into the downslope aquatic environments, which
inevitably changes the proportion of AL vs. PL contributions to stream DOM. Thus, it
is worth to trace chemical change of PL leachates along the stream. The PL leachate
contained only 1345 formulas unique to the PL leachate in comparison to the AL,
accounting for 14% of total assigned formulas (Table 2). Most PL-specific formulas
were more biolabile components, e.g. aliphatics and peptides (73%), followed by highly
unsaturated formulas (23.6%) and aromatics (1.9%). At the downstream site (i.e., Q-
17), 90% of these PL-specific molecular formulas were lost, substantially higher than
that of AL-specific formulas (59%). Furthermore, the vast majority of PL-specific
formulas were lost within < 1 km (Q-1: 83%) whereas only 17% of AL-specific
formulas were lost by Q-1 (Table 2). Therefore, the FT-ICR MS data demonstrate that
permafrost thaw can trigger a rapid degradation of old carbon that was frozen in soils
for thousands of years (Fig. 3a). This is consistent with findings in Arctic fluvial
networks that show the utilization of ancient permafrost carbon in headstream waters
was rapid (Mann et al., 2015; Frey et al., 2016). Therefore, we hypothesize that with
enhanced leaching of deep soil C under continued warming on the QTP, DOM in alpine
streams will be more enriched in biolabile aliphatics/peptides and depleted in
photolabile aromatics
Finally, despite substantial in-stream degradation, some old permafrost-derived
carbon (i.e., polyphenols and highly unsaturated compounds) could persist downstream.
In addition, $CO_2$ produced by respiration of old DOC could be utilized by stream algae
to biosynthesize new DOM with an old carbon age. These effects resulted in a sustained
deviation from modern $^{14}C$-DOC age in the alpine stream (e.g., 160 years BP at Q-19),
and were even detected in large rivers on the QTP (e.g., Yangtze River and Yellow River;



Qu et al., 2017). Thus, under continued warming, a greater quantity of older C may be
transported into large watersheds on the QTP, and thereby exert an important role in
biogeochemical cycles there since older carbon has different photo and bio-lability from
young carbon in AL soils.

**5. CONCLUSIONS**

Permafrost thaw represents positive feedbacks to climate change, but its carbon

alteration and removal mechanism is not well known, particularly for the alpine
permafrost such as the QTP. Here we use multiple analytical methods (e.g., FT-ICR MS,
radiocarbon and UV-visible spectroscopy) to trace spatial and temporal variability of
permafrost DOM along an alpine stream in the northeastern QTP, from which four
conclusions have been reached.

1)   Presently, the AL is the major source to stream DOM with relatively high

aromaticity. This character, combined with strong solar radiation on the QTP, suggests
sunlight may be an important driver for DOM removal in alpine fluvial networks, which
was corroborated by an almost 60% loss of AL specific formulas from the thermo-
erosion gully head to downstream waters.

2)   From summer to fall (seasonal permafrost thawing to freezing cycle), the

concentrations and chemical composition of stream DOM varied significantly at the
thermo-erosion gully head. Even though the total amount of the leached DOC decreased,
the contribution of deep permafrost carbon increased, reflected by an increase of $^{14}$C-
DOC age and a decrease in aromaticity of DOM.

3)   Although both the AL and PL leachate DOM underwent rapid degradation in

the alpine stream, some components with old $^{14}$C-DOC age (mainly highly unsaturated)
were recalcitrant to degradation and could be transported downstream, causing $^{14}$C-



DOC values that were more depleted than modern radiocarbon age downstream in our
study, and even in large watersheds as observed in Qu et al. (2017).
4) With deepening of the AL under continued climate warming on the QTP,
currently stable PL soils will thaw and release greater amounts of old, aliphatic/peptide-
rich DOM to downstream waters. This change in source and chemical composition will
make microbial degradation more important for carbon removal and may shift
downstream microbial communities, even in large watershed systems. All these factors
should be taken into account when interpreting alpine permafrost carbon dynamics
under the amplified climate warming trend on the QTP.

**ACKNOWLEDGEMENTS**
This work was financially supported by the National Basic Research Program of
China (2014CB954001). Y.W. thanks the China Scholarship Council for supporting
study in the United States of America as a joint Ph. D. student. We thank Futing Liu,
Yanyan Yan, Shangzhe Zhou, Xinyu Zhang for assistance in the field. FT-ICR MS was
supported by NSF (DMR-1157490).

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



**Figure and table captions**

**Fig. 1.** Location of the QTP and sampling sites Q1 to Q20. Sites marked by a star were selected for FT-ICR MS and $^{14}$C-DOC analyses. The AL and PL denote the sampling locations of the active and permafrost layers. The blue line and the red line represent the first order and second order stream, respectively, and the blue dashed line denotes stream without GPS data.

**Fig. 2.** (a) Stream water discharge, (b) pH, and (c) conductivity at the sampling sites in 2015; and (d) DOC concentration in stream water and PL/AL leachates collected in 2015 (filled circles) and 2016 (open circles).

**Fig. 3.** UV-visible optical indices of the stream water and PL/AL leachate samples collected in 2015 (filled circles) and 2016 (open circles) on the QTP: $S_{275-295}$ (a) and $SUVA_{254}$ (b).

**Fig. 4.** Variations of $^{14}$C-DOC age across the alpine stream spatially (a), and at headstream Q-1 temporally (b).

**Fig. 5.** van Krevelen diagrams of AL leachate DOM (a), PL leachate DOM (b), headstream DOM Q-1 (c), downstream DOM Q-17 (d), the relative abundance of defined compound class in different samples (e). CA = condensed aromatics, Pol. = polyphenols, Uns. = highly unsaturated compounds, Ali. = aliphatics, Pep. = peptides; and Sug. = Sugar.

**Fig. 6.** (a) Relative contribution of AL leachate DOM to headstream DOM (Q-1); and (b) temporal variation of the DOC concentration at headstream Q-1.

**Table 1** The number of molecular formulas assigned, modified aromaticity index (AI$_{mod}$), mean molecular weight (mean MW) and relative abundance of defined compound classes detected by FT-ICR MS for DOM samples from the QTP, including soil leachates (AL and PL) and stream waters (Q-1, Q-9 and Q-17). CA = condensed aromatics, Pol. = polyphenols, Uns. = highly unsaturated compounds, Ali. = aliphatic, Pep. = peptides.

**Table 2** The number of specific molecules identified in the AL leachate DOM and the PL leachate DOM within the fluvial network, and the percentage of peaks totally degraded during the transportation.



Fig. 1

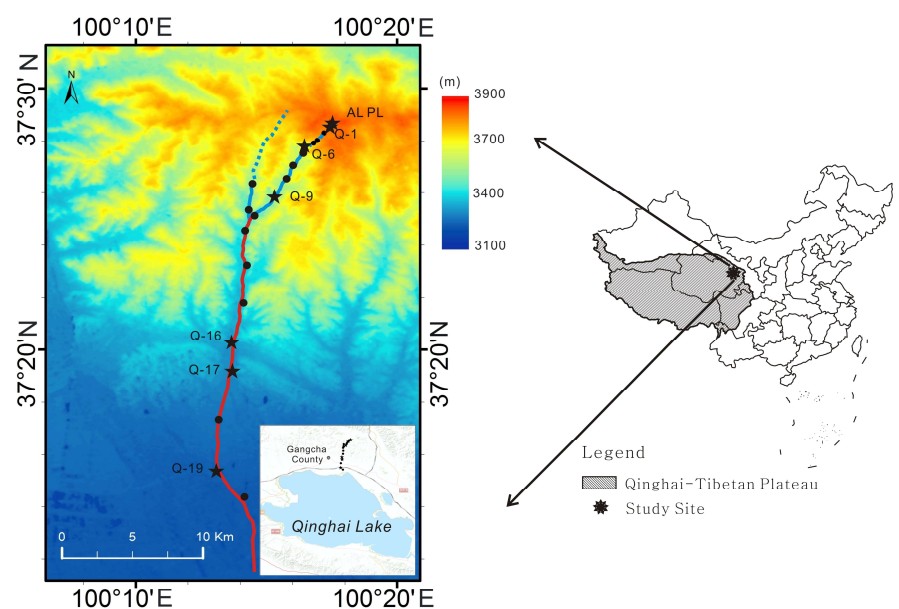







Fig. 2

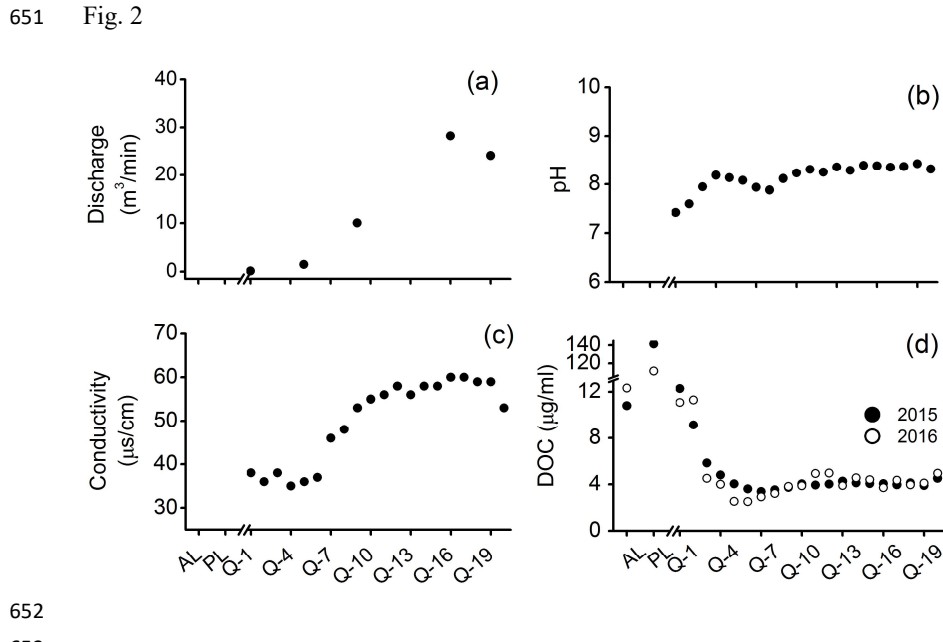







Fig. 3

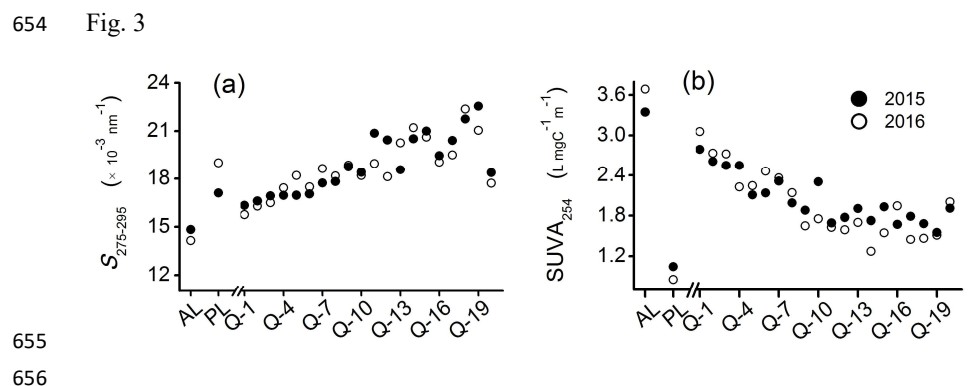







Fig. 4

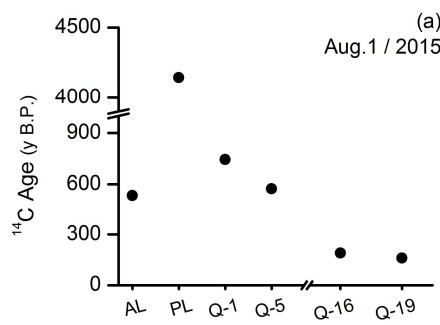
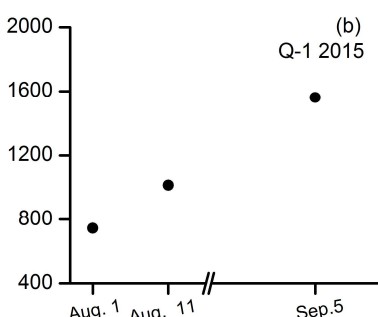





Fig. 5

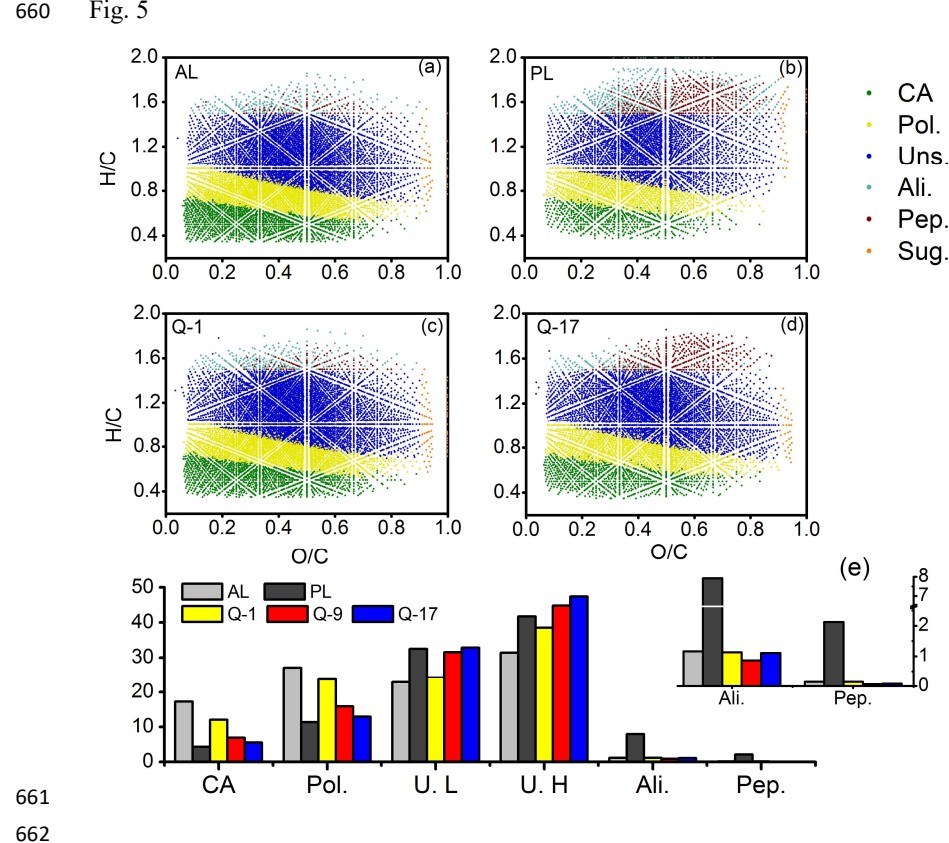





Fig. 6

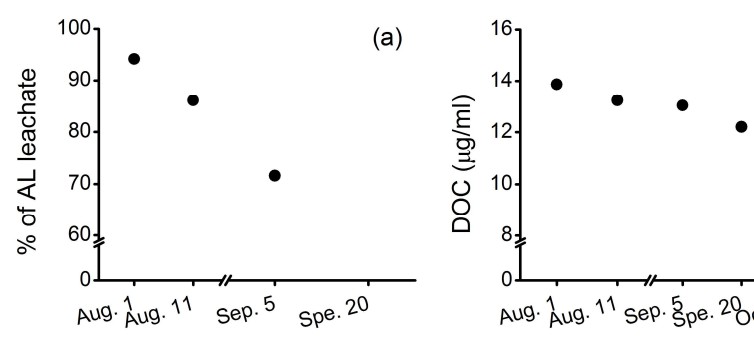





**Table 1.** The number of molecular formulas assigned, modified aromaticity index
(AI$_{mod}$), mean molecular weight (mean MW) and relative abundance of defined
compound classes detected by FT-ICR MS for DOM samples from the QTP, including
soil leachates (AL and PL) and stream waters (Q-1, Q-9 and Q-17). CA = condensed
aromatics, Pol. = polyphenols, Uns. = highly unsaturated compounds, Ali. = aliphatic,
Pep. = peptides.

| Sample | Formulas assigned | Mean MW | AI$_{mod}$ | CA (%) | Pol. (%) | Uns. (%) | Ali. (%) | Pep. (%) |
|---|---|---|---|---|---|---|---|---|
| AL | 14709 | 498.81 | 0.47 | 17.23 | 27.10 | 54.28 | 1.16 | 0.14 |
| PL | 9645 | 452.73 | 0.30 | 4.32 | 11.33 | 74.23 | 7.92 | 2.12 |
| Q-1 | 14924 | 510.07 | 0.43 | 12.05 | 23.69 | 62.85 | 1.14 | 0.14 |
| Q-9 | 11724 | 500.19 | 0.38 | 6.86 | 15.82 | 76.32 | 0.86 | 0.06 |
| Q-17 | 11074 | 486.50 | 0.36 | 5.53 | 12.91 | 80.31 | 1.11 | 0.08 |







**Table 2:** The number of specific molecules identified in the AL leachate DOM and the PL leachate DOM within the fluvial network, and the percentage of peaks totally degraded during the transportation.

| Samples | | All peaks | Only CHO | Contains N | Contains S | Contains N&S | Condensed aromatics | Polyhoenols | Unsatuated | Aliphatics | Peptides |
|---|---|---|---|---|---|---|---|---|---|---|---|
| | AL | 6409 | 1793 | 3370 | 424 | 822 | 1620 | 1720 | 2970 | 38 | 23 |
| AL specific | Q-1 | 5311 (17%) | 1653 (8%) | 2791 (17%) | 349 (18%) | 517 (37%) | 1278 (21%) | 1416 (18%) | 2549 (14%) | 20 (47%) | 14 (39%) |
| | Q-9 | 3365 (47%) | 1294 (28%) | 1917 (43%) | 153 (64%) | 0 (100%) | 748 (54%) | 838 (51%) | 1759 (41%) | 6 (84%) | 1 (96%) |
| | Q-17 | 2623 (59%) | 985 (45%) | 1570 (53%) | 67 (84%) | 0 (100%) | 550 (66%) | 602 (65%) | 1453 (51%) | 5 (87%) | 0 (100%) |
| | PL | 1345 | 515 | 551 | 278 | 0 | 2 | 23 | 318 | 597 | 385 |
| PL specific | Q-1 | 222 (83%) | 90 (83%) | 102 (81%) | 30 (89%) | 0 | 0 (100%) | 11 (52%) | 126 (60%) | 46 (92%) | 36 (91%) |
| | Q-9 | 117 (91%) | 44 (91%) | 46 (92%) | 27 (90%) | 0 | 2 (0%) | 14 (39%) | 96 (70%) | 1 (100%) | 4 (99%) |
| | Q-17 | 130 (90%) | 47 (91%) | 55 (90%) | 28 (90%) | 0 | 2 (0%) | 13 (43%) | 104 (67%) | 6 (99%) | 5 (99%) |