# Peer review of "Spatiotemporal transformation of dissolved organic matter along an alpine stream flowpath on the Qinghai-Tibetan Plateau: importance of source and permafrost degradation"

_Biogeosciences, 2018_

## Referee Comment (RC1) · R. Jaffe (Referee) · 6 Jun 2018

R. Jaffe (Referee)

jaffer@fiu.edu

General Comments:

The abovementioned manuscript describes research on the effect of climate change on permafrost degradation in the Tibetan Plateau and its potential impact on associated fluvial systems, in particular on the dynamics of dissolved organic matter. This research is of global significance as little is known about permafrost degradation in areas other than the arctic, and nearly 70% of alpine permafrost is located in the geographical area

of this study. The research team is composed of highly qualified scientists with ample experience and expertise in the specific field of study, and applying ideal methodologies to reach the outlines objectives of this research initiative. The manuscript is well written, and the data properly presented. The literature is also properly reviewed and well represented. As such, this manuscript is well-suited for the journal Biogeosciences and I recommend it to be published. However, some aspects of the manuscript should be improved prior to acceptance. For example, seasonal variability observed needs to be fully explained; explanations regarding the observed differences in DOM leachate composition between the AL vs PL needs to be better explained; discussion on in-stream generation of DOM through microbial primary productivity should be enhanced and variations along the sampling transect better described; etc. These pending issues are described in more detail below.

Specific Comments:

1) L43: "...in-stream metabolism...": Throughout the manuscript make sure DOM degradation via molecular transformations vs mineralization to CO2 is specified as needed. Similarly, dilution (concentration decrease) vs. 'dilution' (change in relative abundance) through mixing with in-stream DOM from microbial PP?

2) L 54: As in #1 – bio- and photo-transformation vs. mineralization? Both?

3) L61-62: Not sure 'hydrologic inputs' is the best way to word this! Please re-phrase.

4) L116: Please indicate distance in Km. This can be deduced from Fig. 1, but would be helpful here for the reader to easily gain a grasp of the spatial extension of the study.

5) L120-124: Please add more details on the methodology used for leachate collection.

6) L124-127+: Please add distances in m or Km as needed.

7) L156-160: Leachate/Water volumes used for the SPE? How did you avoid break-through?

8) L180: 'Freeze-dried retentates'? Meaning SPE-DOM? Explain or rephrase accordingly.

9) L206-207: Does that mean the in-stream microbial generation of DOM is negligible?

10) L213-216: I do not see any detailed discussion on this inter-annual variability. Please add.

11) L243: Please expand on the discussion of these differences in chemical composition between AL and PL leachates. The information shown in the discussion is highly selective to age and very limited with regards to molecular composition and optical properties. In the first paragraph on page 11 there is some discussion on this with regards to sample Q-1, but nothing much else (i.e. along the sampling transect).

12) L256: How were STDs obtained from n=2?

13) L260: Remove the '(' before 'and'

14) L261-264: idem as above – explain differences in composition between AL and PL.

15) L280: What about seasonal variations in the optical properties and MS data? Missing important information here. Please add.

16) L285-288: This statement seems to make sense, but at the same time the DOC concentration from PL is significantly more elevated compared to AL. How much is 'percolation' due to freezing reduced?

17) Section 4.2: I encourage the authors to actually calculate physical dilution to see if it indeed agrees with the estimation determined based on age variation. Mineralization and in-stream contributions could be roughly estimated by difference based on dilution only.

18) L304-314: Not clear why the authors make comparisons with values observed in coastal systems. Seems irrelevant in this case.

19) L314-318: The size-reactivity continuum (Amon and Benner, 1996) applies well for marine systems. However, it is controversial for terrestrial systems as both similar and opposite trends have been reported in the literature. Considering this, I would focus on the photo-degradation process, which is more likely dominant in this case.

20) L346-353: I would like to see an effort by the authors in enhancing the interpretation of the MS data here. Can molecular formulas generated/added along the transect through microbial in-stream activity be identified? What about photo-transformation products? I assume not all photo-degraded DOM is mineralized to $CO_2$.

21) L392-393: This seems to make sense, but is still mainly speculative. Can you find partial evidence for this from your MS data (i.e. in-stream DOM)? Not sure it is possible.

22) L413: As above – seasonal variations discussion needs to be enhanced.

23) Figure 5: Color code 'dots' are VERY hard to see. Please enlarge accordingly.

---

## Referee Comment (RC2) · Anonymous Referee #1 · 11 Jul 2018

General comments: The manuscript describes the use of multi techniques (UV, AMS, and FT-ICR MS) to characterize dissolved organic matter (DOM) and discuss the temporal and spatial transformation of DOM along an alpine stream. The methodologies are state-of-the-art, the study area will be of interest to readership, the discussion is appropriate, and the manuscript is well-written. There are some issues, listed below, to address prior to publication.

Specific comments:

Figure 5 was not mentioned in the context, which is my most concern. The "highly unsaturated compounds" was classified into L and H, how to define the classification?

From the distributions of H/C and O/C ratios shown in the van Krevelen diagrams in Figure 5, I trust the assignment of compounds is not credible. The authors directly used the data processing results from the software (EnviroOrg), however, the molecule assignment based on accurate mass value cannot guarantee a correct identification on the mass peak, especially on the high mass end. Most "compounds" classified into the CA area (O/C<0.2) should be incorrect assignment of mass peaks with high mass values. This does not means the data processing and the discussion are wrong, in fact, most published papers in the past years have this problem.

In Table 1 and Table 2, as well as many places in the context, the number of assigned molecular formulas was used to discuss the composition of DOM and the degradation along the stream. This is not rigorous and could lead to misunderstanding for readers both on the mass spectrometry analysis and the environmental interpretation. Not like most other analysis instrument detectors, the limit of detection (LOD) of ESI FT-ICR MS for DOM analysis is uncertain, it partially determined by the most abundant peak in the spectrum. Briefly, less mass peaks does not means the composition of the sample is "simple" and compounds not detected in the sample does not means these compounds must in lower concentration.

I suggest the authors provide some raw mass spectrometry data in the supporting information, such as the broad bound mass spectra, expanded mass scale mass spectra.

Line 220. The numbers include the isotope formulae?

Line 221. Elemental analysis was not mentioned in the context.

Line 232. I don't think it is a good manner to compare the changes in MW and AI with percentage values.

---

## Author Comment (AC1) · 25 Jul 2018

On behalf of my coauthors, I really appreciate Dr. Jaffe to supply his comments on our manuscript. We found most of his comments are reasonable,so we accepted them and made corrections in the revised manuscript. The follows are our responses point by point. I also marked all changes in the revised manuscript and submit related figures and text as attachments.

General Comments: The abovementioned manuscript describes research on the effect

of climate change on permafrost degradation in the Tibetan Plateau and its potential impact on associated fluvial systems, in particular on the dynamics of dissolved organic matter. This research is of global significance as little is known about permafrost degradation in areas other than the arctic, and nearly 70% of alpine permafrost is located in the geographical area of this study. The research team is composed of highly qualified scientists with ample experience and expertise in the specific field of study, and applying ideal methodologies to reach the outlines objectives of this research initiative. The manuscript is well written, and the data properly presented. The literature is also properly reviewed and well represented. As such, this manuscript is well-suited for the journal Biogeosciences and I recommend it to be published. However, some aspects of the manuscript should be improved prior to acceptance. For example, seasonal variability observed needs to be fully explained; explanations regarding the observed differences in DOM leachate composition between the AL vs PL needs to be better explained; discussion on instream generation of DOM through microbial primary productivity should be enhanced and variations along the sampling transect better described; etc. These pending issues are described in more detail below.

Specific Comments: 1) L43: ": in-stream metabolism:": Throughout the manuscript make sure DOM degradation via molecular transformations vs mineralization to $CO_2$ is specified as needed. Similarly, dilution (concentration decrease) vs. 'dilution' (change in relative abundance) through mixing with in-stream DOM from microbial PP?

Response: This is a good comment. Since the DOM degradation process is very complex. Besides different types of degradation (photodegradation vs. biodegradation), DOM can be also completely degraded into $CO_2$ or partially degraded to other compounds. For the former, we prefer to call "transformation", while for the latter, we used "mineralization", although in many literatures, "degradation" was simply used for expressing DOM change. In the revised manuscript, we clarify this difference. From line 41-43, we rewrote as "Our study thus demonstrates that hydrological conditions impact the mobilization of permafrost carbon in an alpine fluvial network, the signature

of which is quickly lost through in-stream mineralization and transformation". As for the dilution effect, we referred it to concentration decease at Line 292-304.

2) L 54: As in #1 – bio- and photo-transformation vs. mineralization? Both?

Response: As we mentioned above, we clarify this point in the revised manuscript. We rewrote the sentence as "When permafrost-derived carbon enters aquatic system, it can be rapidly mineralized and transformed by microbes and light" in line 55.

3) L61-62: Not sure 'hydrologic inputs' is the best way to word this! Please re-phrase.

Response: We use 'hydrologic condition' to replace 'hydrologic inputs'.

4) L116: Please indicate distance in Km. This can be deduced from Fig. 1, but would be helpful here for the reader to easily gain a grasp of the spatial extension of the study.

Response: That's a good suggestion. we suppled this information in the revised manuscript. In line 117-118, we wrote as 'The water in the gully flows southward across the hillslope before draining into Qinghai Lake, and the total length of the stream is around 40 km (Fig. 1).'

5) L120-124: Please add more details on the methodology used for leachate collection.

Response: We already added the more detailed description of the sampling method in the revised manuscript (Line 122-125). We wrote as 'At each sampling time, both AL and PL leachates were collected at the depth of 60 cm and 220 cm, respectively, of the gullies' head. 20 L HDPE carboys were cleaned by pure water, 0.1 N hydrochloric acid and pure water prior to use. It usually took 2 days to gather > 15 L leaching waters. After that, the leachate samples were immediately kept on ice and in the dark by aluminum foil. They were transported to the temporary laboratory in the Gangcha County with six hours.'

6) L124-127+: Please add distances in m or Km as needed.

Response: We added this content in the revised manuscript, which is 8.5 km long for

the first order stream and 6.9 km long for another order stream.

7) L156-160: Leachate/Water volumes used for the SPE? How did you avoid break-through?

Response: We actually realized this point. Before SPE, we estimated the maximum volume before loading samples based on the SPE recovery (60% in our case) and the final eluate concentration 40 $\mu$g C/ml. The exact loading volumes vary among samples, but the eluate concentration is similar that might help reduce the selective ionization. In the revised manuscript, we added detailed information on this issue. From line 160 to 166, we wrote as 'They were solid-phase extracted (SPE) using the Bond Elut PPL (Agilent Technologies, 100 mg PPL in 3 ml cartridge), following the procedures of Dittmar et al. (2008). In order to avoid overloading of the SPE column, the aliquot volume of SPE DOM was calculated based on an average SPE recovery (60% for permafrost DOM; Ward, et al., 2015) and a final eluate concentration of 40 $\mu$g C/ml (in ca. 2 ml methanol)." We also cited a reference 'Ward, C. P. and Cory, R. M.(2015) Chemical composition of dissolved organic matter draining permafrost soils. Geochimica Et Cosmochimica Acta. 167, 63-79.'.

8) L180: 'Freeze-dried retentates'? Meaning SPE-DOM? Explain or rephrase accordingly.

Response: We already changed into 'Freeze-dried retentates from ultrafiltration'.

9) L206-207: Does that mean the in-stream microbial generation of DOM is negligible?

Response: In this section we use optical properties to show DOM characteristics, in that way we could quick screen the inter-annual variation between year 2015 and 2016. The lack of inter-annual change did not mean insignificant microbial generation of DOM in stream. Actually, from headwater to downstream water, we observed apparent change in optical parameters of DOM, suggesting substantial transform of DOM by photo or bio-degradation. We discussed this point in section 3.3 Spatiotemporal

change of 14C-DOC age through fluvial networks.

10) L213-216: I do not see any detailed discussion on this inter-annual variability. Please add.

Response: We added the discussion on inter-annual variability in the revised manuscript. From line 204-211 as 'Paired t-test based on S275-295 and SUVA254 of water samples showed no significant inter-annual variation between year 2015 and 2016 (p = 0.716 and p = 0.321, respectively). The mean S275-295 of 2015 and 2016 samples was (14.5 $\pm$ 0.48) $\times$ 10-3 nm-1 for the AL leachates and (18.3 $\pm$ 1.3) $\times$ 10-3 nm-1 for the PL leachates. In the stream waters, the S275-295 ranged from 15.8$\times$ 10-3 to 22.5 $\times$ 10-3 nm-1, increasing in downstream reaches." In the stream waters, the S275-295 ranged from 15.8$\times$ 10-3 to 22.5 $\times$ 10-3 nm-1, increasing in downstream reaches. Mean SUVA254 was 3.52 $\pm$ 0.24 L mg C-1 m-1 for the AL leachates and 0.95 $\pm$ 0.14 L mg C-1 m-1 for the PL leachates, and decreased in the stream from Q-1 to Q-10 (3.06 to 1.27 L mg C-1 m-1), and then remained low (Fig. 3). ', but for the radiocarbon age of the DOM, actually we did not do inter-annual analysis, here we discussed just temporally change in different months in 2015 as showed in line 219-221.

11) L243: Please expand on the discussion of these differences in chemical composition between AL and PL leachates. The information shown in the discussion is highly selective to age and very limited with regards to molecular composition and optical properties. In the first paragraph on page 11 there is some discussion on this with regards to sample Q-1, but nothing much else (i.e. along the sampling transect).

Response: This is good comment. Actually, we have addressed this issue previously. Please see Wang, et al., 2018, Selective leaching of dissolved organic matter from alpine permafrost soils on the Qinghai-Tibetan Plateau. J. Geophys. Res. Biogeosci., 123, 1005-1016, doi: 10.1002/2017jg004343. In this article, we examined and compared the chemical composition of DOM leached from AL and PL. We found the selective leaching in Permafrost soils that upper AL leachates are enriched in aromatic components, whereas deep PL leachates are enriched in alkyl components. In current work, we focus on instream processes of DOM rather than leaching process from soil to headwater. Nevertheless, we added some sentences (line 249-252) as "This difference is likely attributed to selective release of aromatic components from upper AL soils and carbohydrate/protein components from deep PL soils during the thawing process which was observed in our previous study (Wang et al., 2018)." We also cite the reference of Want et al. (2018) in the revised manuscript.

12) L256: How were STDs obtained from n=2?

Response: We are sorry for this mistake. Here we calculated the average value and the average deviation based on two samples. In the revised manuscript, we corrected all the calculated data throughout the manuscript, and here rewrote as "The mean DOC concentration of the AL leachate based on samples from 2015 and 2016 (11.57 $\pm$ 0.77mg/L) is similar to that of the headstream (Q-1; ca. 11.69 $\pm$ 0.60 mg/L), but substantially lower than that of the PL leachates (126.40 $\pm$ 14.80 mg/L), supporting a predominance of AL-leachate DOM in stream waters. In addition, the SUVA254 is 3.52 $\pm$ 0.17 L mg C-1 m-1 for AL leachates and 0.95 $\pm$ 0.10 L mg C-1 m-1 for PL leachates, whereas the S275-295 is (14.49 $\pm$ 0.34) $\times$ 10-3 nm-1 for AL leachates and (18.05 $\pm$ 0.94 ) $\times$ 10-3 nm-1 for PL leachates".

13) L260: Remove the '(' before 'and'

Response: we deleted "(".

14) L261-264: idem as above – explain differences in composition between AL and PL.

Response: We have added some brief information about AL and PL leachates at Line 267-269, but as mentioned above (response to comment 11), we did not give much detailed information in this study.

15) L280: What about seasonal variations in the optical properties and MS data? Missing important information here. Please add.

Response: It is a pity that we did not conduct FT-ICR MS analysis for seasonal samples. But a seasonal variation of DOM could be revealed by our optical analyses. In the revised manuscript, we added the sentence as 'Our result also shows seasonal variations in 14C age and optical parameters of headstream DOM. From summer to fall, the SUVA254 of stream DOM at Q-1 decreased from 2.79 to 2.36 mg C-1 m-1, whereas the S275-295 increased from 16.33 × 10-3 to 16.96 × 10-3 nm-1. These temporal changes indicated that the proportion of aromatic components and high molecular weight compounds decreased with the deepening of permafrost thawing.' Please see the details from line 292 to 297.

16) L285-288: This statement seems to make sense, but at the same time the DOC concentration from PL is significantly more elevated compared to AL. How much is 'percolation' due to freezing reduced?

Response: We agree it would be helpful to distinguish leaching and percolate if we could separate them. Unfortunately, it is very difficult to monitor percolation in fieldwork. So we just separate the whole soil profile into active layer and permafrost layer and discussed combined effects by collecting leaching waters at the Q-1. Nevertheless, several lines of evidence from optical, DOC concentration and FT-ICRMS support our statement that active layer is a major contributor to leachate DOM.

17) Section 4.2: I encourage the authors to actually calculate physical dilution to see if it indeed agrees with the estimation determined based on age variation. Mineralization and in-stream contributions could be roughly estimated by difference based on dilution only.

Response: This is a good comment. We qualitatively discussed the dilution effect in line 305-312. Several lines of evidence from DOC concentration, total water discharge and water conductivity all supported the existence of dilution effect in downstream waters. However, it is difficult to quantify this effect because the lack of DOC and water flux

data of tributaries and groundwater. We may conduct more comprehensive survey in next year and address this issue in future. In current study, we circumvent this problem by tracing unique peaks of DOM by using FT-ICRMS. If these unique peaks disappear along the stream, it suggests the occurrence of biodegradation or photodegradation for the specific type of compounds.

18) L304-314: Not clear why the authors make comparisons with values observed in coastal systems. Seems irrelevant in this case.

Response: We accepted this suggestion and removed related contents in the revised manuscript.

19) L314-318: The size-reactivity continuum (Amon and Benner, 1996) applies well for marine systems. However, it is controversial for terrestrial systems as both similar and opposite trends have been reported in the literature. Considering this, I would focus on the photo-degradation process, which is more likely dominant in this case.

Response: We agree with this suggestion and deleted these sentences. In line 343-344, we rewrote the sentence as 'A strong negative correlation between $S_{275-295}$ and $SUVA_{254}$ ($R^2 = 0.73$, $p < 0.01$) indicates that photodegradation of high molecular weight aromatic compounds (like lignin) may play a role in the decrease of mean molecular weight of DOM along the stream, despite that microbial degradation might also contribute the molecular modification in stream to less extent.'

20) L346-353: I would like to see an effort by the authors in enhancing the interpretation of the MS data here. Can molecular formulas generated/added along the transect through microbial in-stream activity be identified? What about photo-transformation products? I assume not all photo-degraded DOM is mineralized to $CO_2$.

Response: Yes, besides the mineralized molecules and new produced molecules, the partial transformations of DOM can also contribute the change in molecular characteristics in stream. This kind of transformations is a result from combined factors such as

microbial degradation, photo degradation, and also new input from base flow and in-stream generation, among others. In the revised manuscript, we have added a supporting figure (Fig. S1) that shows the change of DOM molecular formula between Q-1 and Q-17, with the decrease of aromatics and the addition of highly unsaturated molecules. From line 363 to 369, we rewrote the sentences as "Concurrent with the rapid loss of AL-specific formulas, some new molecular formulas were detected by FT-ICR MS, which was mainly attributed to in-situ production by stream algae/microbes, and import from groundwater and molecular transformation of leachate DOM. The van Krevelen diagram showed that the new products were mainly composed of highly unsaturated molecules (Fig. S1). The addition of new molecular formulas was also reflected by the 14C enrichment in middle and lower-stream (Fig. 3b)."

21) L392-393: This seems to make sense, but is still mainly speculative. Can you find partial evidence for this from your MS data (i.e. in-stream DOM)? Not sure it is possible.

Response: We identified some new formulas which give some evidence for in-stream production of new DOM, but as mentioned above, these new compounds could be also partially transformed from leachate DOM from bio-, and photo-degradation. In order to distinguish the different pathways, we are currently doing a series of incubation experiments in the laboratory, and wish we can publish those data soon..

22) L413: As above – seasonal variations discussion needs to be enhanced.

Response: We have added the discussion about seasonal changes in the revised manuscript. Please see our response to comment 15.

23) Figure 5: Color code 'dots' are VERY hard to see. Please enlarge accordingly.

Response: we have changed the figure legends and provide a new figure 5 in the revised manuscript.

Please also note the supplement to this comment:

https://www.biogeosciences-discuss.net/bg-2018-182/bg-2018-182-AC1-supplement.zip

[Figure]

Fig. 1.

[Figure]

**Fig. 2.**

---

## Author Comment (AC2) · 27 Jul 2018

We appreciate the anonymous reviewer to supply valuable comments. We accepted most of those comments and made changes in the revised manuscript. The follows are our point by point response, and the changes are marked in green in the revised manuscript. Please see the attached file for details.

General comments: The manuscript describes the use of multi techniques (UV, AMS, and FT-ICR MS) to characterize dissolved organic matter (DOM) and discuss the temporal and spatial transformation of DOM along an alpine stream. The methodologies are state-of-the-art, the study area will be of interest to readership, the discussion is appropriate, and the manuscript is well-written. There are some issues, listed below, to address prior to publication.

Specific comments: Comment 1: Figure 5 was not mentioned in the context, which is my most concern. The "highly unsaturated compounds" was classified into L and H, how to define the classification?

Response: The reviewer is correct here. In our manuscript there is no distinction of between the unsaturated low oxygen (O/C < 0.5) from the unsaturated high oxygen (O/C > 0.5), throughout the manuscript both high and low oxygen are grouped together. So, we have grouped the 'U.H' and 'U.L' together in figure 5 and provides a new figure 5e in the revised manuscript.

Comment 2: From the distributions of H/C and O/C ratios shown in the van Krevelen diagrams in Figure 5, I trust the assignment of compounds is not credible. The authors directly used the data processing results from the software (EnviroOrg), however, the molecule assignment based on accurate mass value cannot guarantee a correct identification on the mass peak, especially on the high mass end.

Response: We somewhat agree with the reviewer. Assignment of molecular formulas in a complex mixture is challenging and this process would be nearly impossible, especially at relatively high m/z as the reviewer points out, if formulas were assigned to individual signals without context. However, in addition to exact mass, the compositional continuum of natural organic matter has been used for the past 15-20 years to assign molecular formulas to natural organic matter (Kujawinski ,E.B., 2002; Stenson, A., et al. 2003). The compositional patterns (homologous series) in natural organic matter enable us to assign molecular formulas to compounds associated with high accurate signals at low m/z where there is only one possible formula. We then extrapolate formula assignment to those at high m/z (in combination with the use of accurate

mass), as many researchers did. We added the references listed above to line 174 of the text, also at lines 513 and 568 as Kujawinski, E.B., (2002). Electrospray Ionization Fourier Transform Ion Cyclotron Resonance Mass Spectrometry (ESI FT-ICR MS): Characterization of Complex Environmental Mixtures. Environ. Forensics 3, 207-216., Stenson, A.C.; Marshall, A.G.; Cooper, W.T., (2003) Exact masses and chemical formulas of individual Suwannee River fulvic acids from ultrahigh resolution electrospray ionization Fourier transform ion cyclotron resonance mass spectra, Anal. Chem. 75, 1275-1284.

Comment 3: Most "compounds" classified into the CA area (O/C<0.2) should be incorrect assignment of mass peaks with high mass values. This does not mean the data processing and the discussion are wrong, in fact, most published papers in the past years have this problem.

Response: There have been some papers that show "islands" or isolate pockets (in van Krevelen space) of formulas that are classified as condensed aromatic compounds. In these cases, we agree with the reviewer that they might be misassigned formulas that are associated with signals of compounds at relatively high m/z. However, this is not the case here. In addition to using homologous series (see above) that is afforded to us by the compositional continuum of natural organic matter (NOM) to assign molecular formulas, we also ensure that the formulas assigned are continuous in carbon number molecular weight, aromaticity, and heteroatom content. The combination of these parameters along with homologous series and accurate mass provide several layers of verification for accurate formula assignments for complex mixtures. The van Krevelen diagrams in Figure 5 show that there are no discontinuities (islands) in the compositional continuum, what's more, the number of condensed aromatic formulas have strong correlation with SUVA 254 (r2 = 0.9679, p<0.01), which all indicated that the formulas are assigned correctly.

Comment 4: In Table 1 and Table 2, as well as many places in the context, the number of assigned molecular formulas was used to discuss the composition of DOM and the

degradation along the stream. This is not rigorous and could lead to misunderstanding for readers both on the mass spectrometry analysis and the environmental interpretation. Not like most other analysis instrument detectors, the limit of detection (LOD) of ESI FT-ICR MS for DOM analysis is uncertain, it partially determined by the most abundant peak in the spectrum. Briefly, less mass peaks do not means the composition of the sample is "simple" and compounds not detected in the sample does not means these compounds must in lower concentration. I suggest the authors provide some raw mass spectrometry data in the supporting information, such as the broad bound mass spectra, expanded mass scale mass spectra.

Response: We agree with the reviewer that interpreting environmental relevance based solely on changes in the number of assigned molecular formulas could be ambiguous. We also agree that changes in the number of assigned formulas is not indicative of compounds at lower concentration as FT-ICR MS is better regarded as a qualitative rather than quantitative method. However, in our study, thousands of specific signals were detected in headwater samples but not detected in downstream waters, providing an unambiguous evidence on compositional change of DOM along the stream. In addition, DOC concentrations of the SPE extracts were normalized for FT-ICR MS analysis. This step largely removed the difference in background matrix between samples. Given these fact, we are confident that the peaks lost in downstream are not caused by the detection limit. In addition, optical and radiocarbon data also showed substantial changes along stream, supporting the changes observed by FT-ICRMS. Nevertheless, we have changed the term "molecular richness" to "chemodiversity" (as described by Kellerman et al. 2014) in lines 227 and 237 for clarity. Frankly speaking, we do not see any necessity of the addition of raw mass spectrum in supporting information. There are over ten thousand peaks in raw mass spectrum for leachate DOM, which reveal little useful information to readers without further data processes. As such, we decided not to do so. But we provided the raw data in the excel sheet in supporting information.

Line 220. The numbers include the isotope formulae?

Response: We did not include the isotopologues here.

Line 221. Elemental analysis was not mentioned in the context.

Response: We changed 'elemental' to 'molecular-level' for clarity.

Line 232. I don't think it is a good manner to compare the changes in MW and AI with percentage values.

Response: We removed the percentages from MW and AI.

Please also note the supplement to this comment:
https://www.biogeosciences-discuss.net/bg-2018-182/bg-2018-182-AC2-supplement.zip

──────────────────────

---

## Referee Report (RR1)

Review of manuscript BG-2018-182 of Wang et al, submitted to Biogeosciences to be published as research article, entitled as "Spatiotemporal transformation of dissolved organic matter along an alpine stream flowpath on the Qinghai-Tibetan Plateau: importance of source and permafrost degradation".

This topic of this MS is alpine permafrost (of the Qinghai-Tibetan Plateau, PRC), which is subject to large scale alteration due to climate change. Especially the deepening of the seasonally thawed surface soil, the active layer (AL), may activate large quantities of frozen soil and connect them to biogeochemical cycles. Of special importance is organic carbon, which can be released by hydrological processes, enter fluvial systems and change regional carbon balance. As such, the topic of the MS is of high relevance to the field and appropriate for the journal Biogeosciences.

Wang and coworker looked at dissolved organic carbon (DOC) sources from two different permafrost soil depths and investigated their fate along a headwater stream and changes within season and years. The authors used a set of complementary methods, optical and mass spectrometric, to characterize DOC quality in their samples.

The main conclusions drawn from their study are that DOC released from the AL contributes most to the stream DOC with seasonal pattern, that this AL derived DOC is rapidly degraded due to in-stream processes and dilution whereas permafrost layer (PL) derived DOC is even faster degraded. Finally the authors use climate predictions of the region to hypothesize about future changes of the alpine stream DOC quality and biogeochemical changes related to this.

This is a re-review of the MS submitted to Biogeosciences and the previous comments by the referees were taken into consideration. The authors of the MS have addressed most of the concerns given by the two previous reviewers and modified the MS accordingly.

However, I have major concerns about the method presentation, the interpretation of the data and the conclusions drawn and cannot recommend publication of this article in its current form.

1. In the study three different types of samples were used, whole water for bulk optical and carbon concentrations, SPE extracts for mass spectrometry and ultrafiltered samples for radiocarbon analysis. Besides a lack of proper description of the methods, no discussion about the potential biases in the interpretation of the results is presented. Especially when combining radiocarbon and mass spectrometry results, this problem becomes evident. In a previous study of the same team of authors (published this year in JGR:Biogeosciences), they acknowledged that ultrafiltration recovers 40% of the total C whereas SPE recovers essentially "the other" 60%. So how should one infer that "the addition of new molecular formulas was also reflected by the 14C enrichment in middle and lower stream" (L362f)? There is a large body of literature dealing with the extraction (biases) of radiocarbon and I encourage the authors to pay more attention to that.

2. The hydrology of the system seems not well considered. E.g. dilution is acknowledged to change the EC and increase the discharge, but this groundwater dilution with a DOC concentration of 3-4 mg/L (L 311ff) is not considered in terms of different DOC quality. In fact, if the discharge increase would be just from GW addition (which is not, as also other tributaries contribute), than the order of magnitude increase in discharge is accompanied by an order of magnitude decrease of DOC concentration between Q-1 and Q16-Q20. How does a plot of discharge vs EC and DOC look like? So when comparing low DOC station Q16-Q20 with high DOC station Q1, an increase of the relative abundance or number of molecular formulas my just be caused by the additional input of GW derived DOC and not by in-stream production of DOM. Similar issues are discussed below.

3. The change of DOC quality along the stream is mainly attributed to photo-degradation, causing depletion of SUVA (aromatics) and increase in spectral slope (molecular weight). The authors argue that due to the high insolation on the QTP, photo-degradation is likely an important process. However, given the high DOC concentration and short water residence time in the headwater (where most changes in DOC concentration and quality occur), make this assumptions at least questionable. Also potential high turbidity of the stream should at least be excluded. But no in-situ light absorption and energy dose was measured, nor was there

an experimental proof of the claimed rapid photo-degradation. In other words, just because in other studies a change in the above mentioned optical properties were observed after experimental UV exposure, not all observed changes in such complex environmental settings can be attributed to the same, single effect.

Detailed comments:

Introduction:
L48ff:  No mentioning about organic carbon in the whole first para, is it not important?
L67ff:  I don't understand this statement. Please explain better why a "space-for-time" approach should reflect seasonal exported permafrost carbon fate. The authors actually attempted to measure the seasonal pattern.
L90:  And later: Please explain how DOM from the PL can actually leach, if the soil is per definition frozen. What are the mobilizations processes?

Materials and Methods:
L122:  If the water really percolates through both layers, than the PL leachate is likely a mixture of AL and PL DOM (as also reflected by the much younger radiocarbon age of the PL leachate as compared to PL bulk soil, see Wang et al. 2018). Were the leachates not filtered?
L137:  The UF filtrate is not mentioned further.
L144:  Given the great detail on DOC measurement, I wonder what the exact method of flow rate determination was?
L165:  This is nicely explained, but did you actually check for recovery? Varying extract DOC concentrations my bias subsequent the FT-ICR measurement mass peak intensities. What was the measurement concentration and was only MeOH used as ESI solvent?
L168:  replace "speed" by "rate"
L169ff: Please report the measurement and evaluation mass range, composition boundaries and applied ppm threshold for formula assignment.
L177ff: How did you derive theses definitions/boundaries of compound "groups"? E.g. a molecule with just 1 N-atom can by definition not be a "peptide" nor is any molecule with > 2 N-atoms automatically a peptide, even if it may by change have the same H/C and /C ratio as a peptide. This unwary and unnecessary use of compound annotations will get the MS community in the same trouble as the EEMS community with their "tryptophan" and "tyrosine" fluorescence. Same applies to polyphenols.
L184:  Not clear which samples were used for radiocarbon analysis

Results:
L199:  Please define AD
L227:  By "molecular chemodiversity" you just mean number of assigned formulas. Really??
L228:  Was the molecular weight calculated as weighted average? And if not, why not?
L234:  "polyunsaturated" was not previously defined.

Discussion:
L250:  Why is this paper by the same team of authors first cited in the discussion? Much of the sampling site and methods has been described already there.
L278:  Please explain equation variables. As no Δ14C values were reported (which I thought was good practice), the equation is of no great use to the reader.
L281ff: More critical than fast degradation (where?) is that also source 14C values may change seasonally and bias the interpretation of changes Al and PL contributions.
L302:  This sentence is inconsistent. When the relative contributions of PL increases, shouldn't that lead to an increase in DOC concentration? Again 14C values may support this, but are apparently lacking? Any support from optical measurements?
L309:  Tributaries also contribute to an increase in discharge.
L341:  Also precipitation, aggregation etc can remove DOC and selectively alter DOC quality.

L343: Also groundwater dilution with different DOC quality can change the relative abundances of peaks. This needs to be taken into account when discussing permafrost DOC quality changes along the stream.

L398: This depletion may only be relative, not absolute, if the amount of AL derived OC stays constant and only the amount of PL derived OC increases.

Conclusions:

L419ff: I'm not convinced that one could state that the loss of Al-specific formulas is an indication for sunlight as driver of DOM removal. (see above discussion)

L430: It was not shown in this study that "components with old 14C-DOC age (…) were recalcitrant to degradation". It was only assessed based on bulk 14C values. Further a direct relation to molecular composition as "highly unsaturated" cannot be established as 14C and MS were measured on different DOC fractions.

Figures and Tables:

Fig.2: Please report DOC concentration in mg/L like in the main text. (also Fig. 6)

Fig.3: I'm confused as in the methods section, sampling of PL and AL was conducted in 2016 and 2017, here data from 2016 and 2015 are displayed.

Table2: Data is presented on a number basis only. How much of the total intensity within each spectrum do the individual groups represent?

---

## Author Response (AR2)

Dear Prof. Steven Bouillon,

Thank you so much for constructive comments. During the past two weeks, we have put together our response to the comments as below. We have also added some minor edits regarding the language of the manuscript to improve ease of reading.

1. In the study three different types of samples were used, whole water for bulk optical and carbon concentrations, SPE extracts for mass spectrometry and ultrafiltered samples for radiocarbon analysis. Besides a lack of proper description of the methods, no discussion about the potential biases in the interpretation of the results is presented. Especially when combining radiocarbon and mass spectrometry results, this problem becomes evident. In a previous study of the same team of authors (published this year in JGR: Biogeosciences), they acknowledged that ultrafiltration recovers 40% of the total C whereas SPE recovers essentially "the other" 60%. So how should one infer that "the addition of new molecular formulas was also reflected by the $^{14}$C enrichment in middle and lower stream" (L362f)? There is a large body of literature dealing with the extraction (biases) of radiocarbon and I encourage the authors to pay more attention to that.

**Response:** We understand the reviewer's concern here. It is true that high molecular weight ultrafiltrated dissolved organic matter, so called HMW UDOM, usually has a younger $^{14}$C age compared to bulk DOM. However, the offset of the $^{14}$C age is relatively constant between UDOM and bulk DOM, suggesting that it is still feasible to assess biogeochemical processes of DOM based on HWM UDOM. Actually, because of low carbon concentrations and high salt abundance, it is routine to conduct pretreatment prior to mass spectral and NMR analyses in the literature (e.g., Chen et al., 2016. Structural and compositional changes of dissolved organic matter upon solid-phase extraction tracked by multiple analytical tools. Anal. Bioanal. Chem. 408, 6249-6258). In our manuscript, we used the $^{14}$C data of HWM UDOM to discuss the relative contribution of permafrost layer (PL) and active layer (AL) leached DOM to the stream at site Q-1 and compared to the $^{14}$C age from headstream to downstream waters. We attempt to trace temporal and spatial changes of DOM along the alpine stream rather than provide an accurate radiocarbon age of bulk DOC at each site. Thus, the extraction biases will not affect our discussion. In order to clarify this point, we have added the reference of "Broek, et al., 2017, Coupled ultrafiltration and solid phase extraction approach for the targeted study of semi-labile high molecular weight and refractory low molecular weight dissolved organic matter, Marine Chemistry, 194, 146-157". We also added the sentence as in line 287-290 as "Broek et al. (2017) found that although the $^{14}$C age of HMW UDOM was significantly younger than that of bulk DOM from North Central Pacific Ocean, the offset between them is constant in the whole marine system. This result suggests that HMW UDOM can serve as a proxy for bulk DOM." We also emphasize the HWM UDOM in several places of the revised manuscript. Please see our resubmission for details. Finally, we also note that the recovery of HMW DOM in freshwaters and the material retained on an SPE-PPL cartridge are not opposing to one another (they do not represent 40% and the other 60% as the reviewer suggests). Likely the vast majority of HMW DOM was collected by

ultrafiltration, and the same HMW fraction was also retained on the SPE-PPL cartridge, then the PPL retains slightly more in addition (see Mopper et al., 2007; Chem. Rev. 2007, 107, 2, 419-442 for a detailed discussion of this topic).

2. The hydrology of the system seems not well considered. E.g. dilution is acknowledged to change the EC and increase the discharge, but this groundwater dilution with a DOC concentration of 3-4 mg/L (L 311ff) is not considered in terms of different DOC quality. In fact, if the discharge increase would be just from GW addition (which is not, as also other tributaries contribute), than the order of magnitude increase in discharge is accompanied by an order of magnitude decrease of DOC concentration between Q-1 and Q16-Q20. How does a plot of discharge vs EC and DOC look like? So, when comparing low DOC station Q16-Q20 with high DOC station Q1, an increase of the relative abundance or number of molecular formulas my just be caused by the additional input of GW derived DOC and not by in-stream production of DOM. Similar issues are discussed below.

**Response:** We did not attribute the increase in the relative abundance or number of molecular formulas to in-stream production only, and we also never say non-existence of groundwater inputs. We mentioned several times in the manuscript about groundwater contribution. For example, 1) In line 329-334, we wrote as "Dilution from groundwater is likely since groundwater discharge sustains baseflow of rivers and streams in the QTP (Ge et al., 2008). Downstream groundwater inputs were further supported by the order of magnitude increase in discharge (1.49 to 24.14 m3/min) and increase in conductivity (37 to 60 μs/cm). Moreover, downstream DOC concentrations remained about 3.0-4.0 mg/L (Q-15 to Q-20), indicative of the low DOC concentrations of groundwater. Conversely, a tributary that originated from another thermo-erosion gully merged into the study stream, however, the different tributaries exhibited similar DOC concentrations (e.g., Q-9 and Q-10 vs. Q-11 and Q-12; Fig. 2d)". 2) In line 377-383, we wrote as "**Concurrent with the rapid loss of AL-specific formulas, some new molecular formulas were detected by FT-ICR MS, which was mainly attributed to in-situ production by stream algae/microbes, an import from groundwater and molecular transformation of leachate DOM. The van Krevelen diagram showed that the new products were mainly composed of highly unsaturated molecules (Fig. S1). The addition of new molecular formulas was also reflected by the $^{14}$C enrichment in middle and lower-stream (Fig. 3b)**". However, strong in-stream processes are apparent because 90% of the PL-specific molecular formulas and 59% of AL-specific formulas were lost, which cannot be explained by addition of groundwater or in-situ contribution. In order to trace in-stream processes, we must find some unique tracers that are specific for source of the DOM. That is the reason we used ultrahigh-resolution FT-ICRMS. Ultimately groundwater addition cannot explain the observed trends as we see loss of molecular formula, thus we attribute this to processing of the DOM not mixing with a new source.

3. The change of DOC quality along the stream is mainly attributed to photo-degradation, causing depletion of SUVA (aromatics) and increase in spectral slope (molecular weight). The authors argue that due to the high insolation on the QTP, photo-degradation is likely an important process. However, given the high DOC concentration and short water residence time in the headwater (where most changes in DOC concentration and quality occur), make this assumptions at least questionable. Also potential high turbidity of the stream should at least be excluded. But no in-situ light absorption and energy dose was measured, nor was there an experimental proof of the claimed rapid photo-degradation. In other words, just because in other studies a change in the above mentioned optical properties were observed after experimental UV exposure, not all observed changes in such complex environmental settings can be attributed to the same, single effect.

**Response:** We understand the reviewers concerns here. Our major objectives are already stated in the introduction section of the manuscript: 1) determine the dominant sources of alpine stream DOM on the QTP (active layer (AL) vs. permafrost layer (PL)), and 2) trace the persistence and degradation of permafrost-derived DOM in an alpine fluvial network (line 90-97). For the first objective, we used DOC concentration, optical and radiocarbon data to compare AL and PL, and all data showed the predominance of AL contribution (see section 4.1). For the second objective, we found that formulas which are unique to AL or PL leachates in the system, so the degradation of permafrost-derived DOM is significant which was also supported by the optical and radiocarbon data. However, the most important source to the stream is permafrost DOM from AL which is also characterized by high aromaticity. Based on these facts, we merely suggest photo-degradation as a potentially important pathway for the removal of DOM in the QTP stream. We do not attempt to quantify the weight of photo and bio-degradation to the removal of DOM as that is beyond the scope of the current manuscript. But we are conducting the laboratory simulation experiments to assess these two degradation pathways. Ultimately our goal is merely to put forward reasonable discussion of the DOM compositional changes observed in the data and so we have toned back the language to make it apparent that we are suggesting photodegradation may be important as opposed to stating that is the case. We sincerely hope that is clearer in the new edited version.

Detailed comments:
Introduction:
L48ff: No mentioning about organic carbon in the whole first para, is it not important?
**Response**: "Carbon" in our manuscript is specifically for "Organic Carbon". In order to clarify this point, we have added "organic" before "Carbon or C" on line 48, 54, 56 in first paragraph and line 93 in the forth paragraph.. Apologies this should have been included.

L67ff: I don't understand this statement. Please explain better why a "space-for-time"

approach should reflect seasonal exported permafrost carbon fate. The authors actually attempted to measure the seasonal pattern.

**Response**: We have rephrased the statement to "Since the persistence of DOM in aquatic systems is related to chemical composition (Kellerman et al., 2015; Kellerman et al., 2018), it is important for chemical characterization of DOM at different spatial and temporal scales". (line 67-72)

L90: And later: Please explain how DOM from the PL can actually leach, if the soil is per definition frozen. What are the mobilizations processes?

**Response**: With the increase temperature on the Qinghai-Tibet Plateau, thermo-erosion and thermokarst occurs widely, resulting in the exposure of deep permafrost layer (PL) to air. Therefore, the leaching from PL is feasible and particularly happens towards the end of the summer (i.e. period of maximal permafrost thaw). We conducted the study along an alpine stream that originates in the thermo-erosion gully and so long frozen material is mobilizing due to increasing temperatures at this location.

Materials and Methods:

L122: If the water really percolates through both layers, than the PL leachate is likely a mixture of AL and PL DOM (as also reflected by the much younger radiocarbon age of the PL leachate as compared to PL bulk soil, see Wang et al. 2018). Were the leachates not filtered?

**Response**: Yes, we filtrated leachate samples before further analysis. We also agree with the reviewer's comment that vertical percolate is possible, reflected by younger $^{14}$C age of PL-leachates compared to that of PL-soil organic matter.

L137: The UF filtrate is not mentioned further.

**Response**: UF filtrate were actually mentioned in 'Radiocarbon analysis' part on line 185 as "Freeze-dried retentates from ultrafiltration were fumigated with concentrated hydrochloric acid…" and in the revised manuscript, we have made clear statement on radiocarbon age of HMW UDOM.

L144: Given the great detail on DOC measurement, I wonder what the exact method of flow rate determination was?

**Response**: We have added the related information in the manuscript as "A portable propeller-type current meter was used to measure the flow rate at the stream cross-section, 5–9 measurements were performed. The water flux was calculated according to average flow rate and cross-sectional area of the stream." (line 147 to 150).

L165: This is nicely explained, but did you actually check for recovery? Varying extract DOC concentrations my bias subsequent the FT-ICR measurement mass peak intensities. What was the measurement concentration and was only MeOH used as ESI solvent?

**Response**: We did not check for recovery here, but Ward et al. (2015) have calculated the extraction recovery of permafrost DOM, and Dittmar reported the recovery for different types of DOM by PPL. So, we think their recovery data is applicable to our samples since we used the same types of DOM source materials and methodology. The measurement concentration is 40 μg C/ml and the solvent was MeOH, we have mentioned that on line 170-171.

L168: replace "speed" by "rate"
**Response**: We accepted this suggestion and replace "speed" with "rate" in line 146 of the revised manuscript.

L169ff: Please report the measurement and evaluation mass range, composition boundaries and applied ppm threshold for formula assignment.
**Response** We added the requested information. From line 172 to 179, we added as "The direct infusion flow rate was 0.7 μL/min. A total of 100 broadband scans between m/z 150-2000 were co-added for each mass spectrum. After internal calibration in MIDAS Predator Analysis (NHMFL), formulas were assigned based on published rules to signals > 6σ RMS baseline noise (Stubbins et al., 2010) using EnviroOrg®™ software (Corilo, 2015) and categorized by compound class based on the elemental composition of molecular formulas (Kujawinski, 2002; Stenson et al. 2003; Spencer et al., 2014). Formulas with mass measurement accuracy < 0.4 ppm were assigned within the following compositional constraints: C1-100, H2-200, O1-30, N0-3, S0-2."

L177ff: How did you derive theses definitions/boundaries of compound "groups"? E.g. a molecule with just 1 N-atom can by definition not be a "peptide" nor is any molecule with > 2 N-atoms automatically a peptide, even if it may by change have the same H/C and /C ratio as a peptide. This unwary and unnecessary use of compound annotations will get the MS community in the same trouble as the EEMS community with their "tryptophan" and "tyrosine" fluorescence. Same applies to polyphenols.
**Response:** We understand the reviewers concern here thus the use of "like" terminology which we have adopted now throughout. This is the same for the fluorescence community, I don't think they are saying it for example is "tryptophan" fluorescence but "tryptophan-like" and thus it fluoresce in the same part of optical space as tryptophan. We understand the reviewers concerns with similar terminology for FT-ICR MS data and we are merely trying to make the data more accessible for the non-specialist. We agree N-containing aliphatic might be more appropriate than "peptide-like" but we clearly define several groups based on molecular composition – the group names are broad and hopefully help the reader assess what's generally present, just like many previous studies do, and this kind of classification is helpful for us and readers to understand the change of DOM compositions. In our study, we used the most widely used classification and current standards.

L184: Not clear which samples were used for radiocarbon analysis

**Response**: This is a good point. We marked the samples for $^{14}$C measurement in figure 4, including spatial distribution of DOM from the sites Q1, Q5, Q16, Q19, AL and PL leachates, as well as temporal variations of DOM from the site Q1.

L199: Please define AD
**Response**: AD is "Average Deviation" and has been clarified.

L227: By "molecular chemodiversity" you just mean number of assigned formulas. Really??
**Response**: Yes, and this is a commonly used term.

L228: Was the molecular weight calculated as weighted average? And if not, why not?
**Response:** We calculated mean molecular weight based on relative abundance of FT-ICR MS signals. We have added this information in the methods on line 186-188 as "The relative abundance of the defined compound class, mean molecular weight and $AI_{mod}$ of each sample were all weighted by relative abundance in each spectrum."

L234: "polyunsaturated" was not previously defined.
**Response**: Here "polyunsaturated" means "highly unsaturated compounds (Uns.)" which is defined as AImod < 0.5, H/C < 1.5 in section 2.3. We have revised this in the manuscript.

Discussion:

L250: Why is this paper by the same team of authors first cited in the discussion? Much of the sampling site and methods has been described already there.
**Response**: We have added the reference of Wang et al. (2017, JGR: Biogeosciences) in the introduction and results sections. On line 78-81 we wrote as "Consequently, the permafrost soils on the QTP have begun to thaw and collapse, causing abundant carbon loss from in-situ degradation (Mu et al., 2016) and relocation (e.g., selective leaching in different soil layers; Wang et al., 2018).", and on line 113-114, we wrote as "detailed description on the collapse can be found in Wang et al. (2018).".

L278: Please explain equation variables. As no $\Delta^{14}$C values were reported (which I thought was good practice), the equation is of no great use to the reader.
**Response**: This is a good comment. In the equation, $f_{AL}$ and $f_{PL}$ are the fraction of AL and PL-derived C in the DOM, and we have added $\Delta^{14}$C values in the results section supplementary table S1 in the revised manuscript.

L281ff: More critical than fast degradation (where?) is that also source $^{14}$C values may change seasonally and bias the interpretation of changes Al and PL contributions.
**Response**: This is a good concern, changes of source $^{14}$C values do change the AL and PL contribution, but in the binary mixing model we used the fixed end-member value

for AL and PL, because these values will not change a lot throughout the whole season, as the depth of the collapse did not change.

L302: This sentence is inconsistent. When the relative contributions of PL increases, shouldn't that lead to an increase in DOC concentration? Again $^{14}$C values may support this, but are apparently lacking? Any support from optical measurements?

**Response**: We don't agree with reviewer on this comment. With the relative contribution of PL increases, although total DOC concentration decreased, the proportion of PL-derived C increased. We already reported increased old carbon contribution and change in optical property during this process. In line 297-304, we wrote as "Headwater $^{14}$C age of HMW UDOM increased from summer to fall (Fig. 4b), reflecting an enhanced contribution of old carbon from the deeper soils (i.e., PL), however, the AL still accounted for $\geqslant$ 72% of total DOC exported (Fig. 6a). This binary mixing model may overestimate the contribution of AL to stream DOC since PL-derived DOC may be degraded faster than AL-derived DOC, due to the high biolability of ancient permafrost carbon as shown in Arctic ecosystems (Vonk et al., 2013). Nonetheless, the AL appears as a major contributor to stream DOC in the QTP." Both support enhanced contribution of PL-derived C from summer to fall.

L309: Tributaries also contribute to an increase in discharge.

**Response**: We agree with this comment. But we already discussed this issue in the manuscript and examined DOM composition of one tributary in the mid-stream. In line 331-337, we wrote: "Conversely, a tributary that originated from another thermo-erosion gully merged into the study stream, however, the different tributaries exhibited similar DOC concentrations (e.g., Q-9 and Q-10 vs. Q-11 and Q-12; Fig. 2d). The similarities in DOC concentrations were attributed to homogeneity in dominant vegetation, soil type and climate, and thus, homogeneity in DOM inputs to the different tributaries in our study area. Therefore, additional tributaries could not explain the spatial pattern of DOC concentration."

L341: Also precipitation, aggregation etc. can remove DOC and selectively alter DOC quality.

**Response**: We understand the reviewer's concern here. It is true that aggregation and precipitation can potentially change the DOC concentration and quality in many cases. However, in our case, due to very steep hillslope, the sediment and floc deposition is insignificant and we did not observe fine sediments in the stream bed. So, in the revised manuscript, we added the sentences as "The DOC concentration decreased (12.48 to 3.13 mg/L) from upper to mid-stream (Q-1 to Q-5), which could potentially be attributed to several reasons, like aggregation/precipitation, dilution effects, and in-stream degradation of DOM. The aggregation/precipitation is likely unimportant in our case because the steep gradient of sampling sites prevents significant sediment and floc deposition on stream bedrocks, although this effect can't be excluded completely." Please see line 321-327 for details.

L343: Also groundwater dilution with different DOC quality can change the relative abundances of peaks. This needs to be taken into account when discussing permafrost DOC quality changes along the stream.

**Response**. This is true, but the loss of unique molecular formulas can't be explained by groundwater dilution. Since the groundwater input can only add molecular diversity, but not reduce the unique PL and AL-formulas. In our study, about 90% of unique PL-formulas and 59% of AL-formulas were lost in mid and lower stream, which strongly suggest active degradation in stream.

L398: This depletion may only be relative, not absolute, if the amount of AL derived OC stays constant and only the amount of PL derived OC increases

**Response**: That's true, it is relative. Here we hypothesize "with enhanced leaching of deep soil C under continued warming on the QTP, DOM in alpine streams will **be more enriched** in biolabile aliphatics/peptide-like groups and depleted in photolabile aromatics". We predict with further climate warming, the deepening of the AL will lead to the relative proportion of biolabile aliphatics/peptide-like groups in leachate DOM to increase whereas the relative proportion of photolabile aromatics in leachate DOM will decrease.

Conclusions:

L419ff: I'm not convinced that one could state that the loss of Al-specific formulas is an indication for sunlight as driver of DOM removal. (see above discussion)

**Response**: We assume our statements have been misinterpreted here, since we did not intend to claim that sunlight was the only process responsible for the loss of Al-specific formulas. The AL-leachate is enriched in aromatic carbon that is photo-labile but relatively resistant to biodegradation. In addition, the Qinghai-Tibet plateau is characterized by strong UV light and low temperature. These characteristics create a favorable condition for photodegradation of aromatic enriched DOM. So, we draw a conclusion that "Presently, the AL is the major source to stream DOM with relatively high aromaticity. This character, combined with strong solar radiation on the QTP, suggests sunlight may be an important driver for DOM removal in alpine fluvial networks, which was corroborated by an almost 60% loss of AL specific formulas from the thermo-erosion gully head to downstream waters." As noted in the comment above we are merely putting forward a legitimate conclusion from the data presented.

L430: It was not shown in this study that "components with old $^{14}$C-DOC age (…) were recalcitrant to degradation". It was only assessed based on bulk 14C values. Further a direct relation to molecular composition as "highly unsaturated" cannot be established as $^{14}$C and MS were measured on different DOC fractions.

**Response**: We understand this concern here, this highly unsaturated component is extracted by SPE, and the $^{14}$C age is from HMW UDOC, so we have identified the specific component as "some components with old $^{14}$C-DOC age were recalcitrant to degradation and could be transported downstream" on line 434.

Figures and Tables:

Fig.2: Please report DOC concentration in mg/L like in the main text. (also Fig. 6)

**Response:** we have made change in the revised manuscript.

Fig.3: I'm confused as in the methods section, sampling of PL and AL was conducted in 2016 and 2017, here data from 2016 and 2015 are displayed.

**Response**: The sampling was conducted in 2015 and 2016 rather than 2016 and 2017. We have made correction in the method section.

Table2: Data is presented on a number basis only. How much of the total intensity within each spectrum do the individual groups represent?

**Response**: In fact, it is impossible to quantify total intensity of peaks by FT-ICR MS. So, we changed the title of table 2 into "The number of specific molecules identified in the AL leachate DOM and the PL leachate DOM within the fluvial network, and the change in the relative abundance of each formula during the transportation".